# Effects of image homogeneity on stenosis visualization at 7 T in a coronary artery phantom study: With and without $B_1$-shimming and parallel transmission

**Stefan Herz**[1,2]*, **Maria R. Stefanescu**[1], **David Lohr**[1], **Patrick Vogel**[3], **Aleksander Kosmala**[1,2], **Maxim Terekhov**[1], **Andreas M. Weng**[2], **Jan-Peter Grunz**[1,2], **Thorsten A. Bley**[2], **Laura M. Schreiber**[1]

**1** Comprehensive Heart Failure Center (CHFC), Chair of Molecular and Cellular Imaging, University Hospital Würzburg, Würzburg, Germany, **2** Department of Diagnostic and Interventional Radiology, University Hospital Würzburg, Würzburg, Germany, **3** Department of Experimental Physics V, University of Würzburg, Würzburg, Germany

* Herz_s@ukw.de

**Data Availability Statement:** All relevant data are within the paper and its Supporting information files.

## Abstract

### Background

To investigate the effects of $B_1$-shimming and radiofrequency (RF) parallel transmission (pTX) on the visualization and quantification of the degree of stenosis in a coronary artery phantom using 7 Tesla (7 T) magnetic resonance imaging (MRI).

### Methods

Stenosis phantoms with different grades of stenosis (0%, 20%, 40%, 60%, 80%, and 100%; 5 mm inner vessel diameter) were produced using 3D printing (clear resin). Phantoms were imaged with four different concentrations of diluted Gd-DOTA representing established arterial concentrations after intravenous injection in humans. Samples were centrally positioned in a thorax phantom of 30 cm diameter filled with a custom-made liquid featuring dielectric properties of muscle tissue. MRI was performed on a 7 T whole-body system. 2D-gradient-echo sequences were acquired with an 8-channel transmit 16-channel receive (8 Tx / 16 Rx) cardiac array prototype coil with and without pTX mode. Measurements were compared to those obtained with identical scan parameters using a commercially available 1 Tx / 16 Rx single transmit coil (sTX). To assess reproducibility, measurements ($n = 15$) were repeated at different horizontal angles with respect to the $B_0$-field.

### Results

$B_1$-shimming and pTX markedly improved flip angle homogeneity across the thorax phantom yielding a distinctly increased signal-to-noise ratio (SNR) averaged over a whole slice relative to non-manipulated RF fields. Images without $B_1$-shimming showed shading artifacts due to local $B_1^+$-field inhomogeneities, which hampered stenosis quantification in severe cases. In contrast, $B_1$-shimming and pTX provided superior image homogeneity.

**Funding:** Parts of this study were funded by the German Ministry of Education and Research (BMBF) with grant 01EO1004 and 01EO1504 (LMS). URL bmbf.de. The funders had no role in study design, data collection and analysis, decision to publish, or preparation of the manuscript.

**Competing interests:** The authors have declared that no competing interests exist.

Compared with a conventional sTX coil higher grade stenoses (60% and 80%) were graded significantly ($p<0.01$) more precise. Mild to moderate grade stenoses did not show significant differences. Overall, SNR was distinctly higher with $B_1$-shimming and pTX than with the conventional sTX coil (inside the stenosis phantoms 14%, outside the phantoms 32%). Both full and half concentration (10.2 mM and 5.1 mM) of a conventional Gd-DOTA dose for humans were equally suitable for stenosis evaluation in this phantom study.

## Conclusions

$B_1$-shimming and pTX at 7 T can distinctly improve image homogeneity and therefore provide considerably more accurate MR image analysis, which is beneficial for imaging of small vessel structures.

## Background

Coronary artery disease (CAD) remains one of the leading causes of mortality in industrialized countries [1]. For the detection of significant stenoses and prognostic classification of patients at risk of cardiovascular events, non-invasive imaging techniques are highly desirable [2]. For this purpose, international guidelines recommend computed tomography coronary angiography as a well-established tool in appropriately selected patients [3, 4]. However, important potential strengths of magnetic resonance coronary angiography (MRCA) have maintained ongoing research interest in this topic. These advantages include imaging free of ionizing radiation, lumen visualization in the presence of calcifications, the potential of detailed tissue characterization, and concomitant non-invasive assessment of myocardial function, perfusion, and myocardial viability [2, 5]. Currently, MRCA has limited indications in clinical practice at 1.5 Tesla (T) and 3 T due to limits in spatial resolution, long scan times, motion artifacts and the partial volume effect [2].

Ultra-high field (UHF, $B_0 \geq 7$ T) cardiac magnetic resonance imaging (CMR) features the potential of a distinctively higher signal-to-noise ratio (SNR) [6–8], increased parallel imaging ability, and longer $T_1$ relaxation time [9–11] compared with standard CMR at 1.5 or 3 T. Hence, UHF-CMR at 7 T promises increased spatial and temporal resolution, shortened acquisition time, and potentially optimized image contrast. In a study comparing *in-vivo* human imaging of the right coronary artery at 3 and 7 T in young, healthy volunteers, parameters related to image quality attained at 7 T equaled or surpassed those from 3 T [12]. Another study showed the feasibility of high resolution bright blood coronary MRA at 7 T [13]. Reiter et al. recently demonstrated that commercially available 7 T MRI systems allow for morphological and functional analysis similar to the clinically established CMR routine approach [14]. However, UHF-CMR remains challenging in particular due to high specific absorption rates (SAR), inhomogeneity of the main field $B_0$ as well as the transmit radiofrequency field $B_1^+$. The inherently short electromagnetic field wavelength of the radiofrequency (RF) of ~300Mhz results in strong spatial inhomogeneity of the transmit $B_1^+$-excitation profile [7, 15], which often leads to inhomogeneous tissue contrast and potential signal loss in target organs such as the heart [16–18].

A promising method to overcome these constraints is parallel transmission (pTX), the use of multiple independent transmission channels, which provides full spatial and temporal control of the RF fields [19]. Multiple coils are used to 'shim' the $B_1^+$ field in an analogous manner

to $B_0$ shimming, i.e. shimming of the static magnetic field [19], thus improving homogeneity or reducing transmitted power and global SAR [20]. It has been shown in several *in-vivo* UHF-MRI brain and cardiac applications that $B_1$-shimming is not only beneficial but actually needed for many applications to obtain acceptable image quality [10, 20–25]. In particular, initial results on the benefit of $B_1$-shimming for in-vivo coronary artery imaging have been shown for the left coronary artery [26].

While it is perfectly known that $B_1$-induced artifacts leading to SNR degradation may cause problems in obtaining quantitative information from 7 T MR images, the impact of these artifacts on the large number of specific medical imaging aims is still widely unexplored.

In this context, a basic phantom study providing information on what could be achieved with sTX and pTX arrays regarding image SNR homogeneity is a useful first step toward broader research applications of 7 T CMR in humans especially concerning improvements in visualization and quantification of vascular stenoses.

In the present study, we demonstrate the capability and limitations of a commercial single transmit coil for stenosis visualization. Furthermore, we aim to assess the vendor-provided platform for $B_1$-shimming using a pTX array prototype, which is now commercially available. The performance of these two different RF coils (both having 16 array elements) driven in different modes (single transmission (sTX) and pTX) is investigated with respect to image quality and the quantification of mild, moderate, and severe stenoses. Additionally, dilution series were performed to determine the optimal contrast agent concentration at 7 T for the assessment of vascular stenosis phantoms.

## Methods

### Stenosis and thorax phantoms

Stenosis phantoms (Fig 1) were designed to emulate different grades of no, mild and high-grade stenosis (0%, 20%, 30%, 40%, 60%, 80% and 100%) using computer-aided design (CAD) software (Inventor 2016, Autodesk Inc., Mill Valley, USA). Phantoms were 3D-printed on a Form2 3D-printer (FormLabs Inc, Somerville, USA; material: Clear Photoreactive Resin). The diameter of the parent vessel measured 5 mm to simulate the dimensions of the left main stem coronary artery [27].

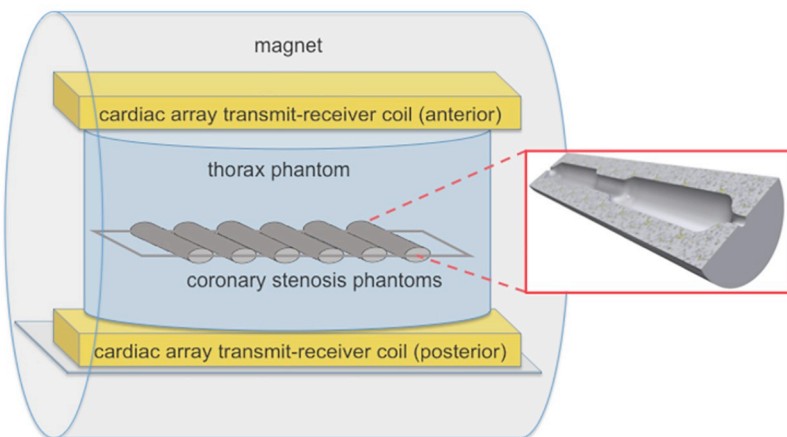

**Fig 1. Side view of the MRI scanner showing the position of the stenosis phantoms in the thorax phantom and the cardiac array transmit-receive coil.** Magnified out in the red box is the cross-section of a stenosis phantom (computer-aided design (CAD)-Model for 3D-printing) with a stenosis of 40% in the center of the vessel phantom.

An oval-shaped thorax phantom was constructed with a maximum inner diameter of 30 cm and a height of 25 cm. The phantom was filled with a liquid featuring dielectric properties of muscle tissue ($\sigma$ = 0.79 S/m, $\varepsilon_r$ = 59), as described in the literature [28]. Stenosis phantoms were fixed with hot glue next to each other in a plastic frame with an approximate inter-phantom distance of 1 cm. The frame was then positioned horizontally in the center of the thorax phantom in a depth of 7 cm below the surface using a 4-point fixation system with suture material.

## Contrast agent

Gd-DOTA (0.5 mmol/ml, Dotarem, Guerbet, Roissy-Charles de Gaulle, France) was used as a contrast agent in all studies. To determine a suitable Gd-DOTA concentration for subsequent $B_1$-shimming experiments, a dilution series was imaged using different contrast agent concentrations (concentration A, B, C; mixture with physiological saline solution) in accordance with previous studies [29, 30]. Physiological saline solution was used as reference (concentration D). Concentration A (10.2 mM) corresponds to a manlike arterial peak concentration after i.v. bolus injection of a single dose of Gd-DOTA (0.2 mmol/kg). Concentration B (5.1 mM) represents an arterial peak concentration after i.v. injection of half a dose. Concentration C (1.0 mM) corresponds to the arterial concentration 1 min after i.v. injection of a single dose. Four imaging series of the stenosis phantoms filled with the different concentrations were obtained and evaluated by two radiologists as described below. Concentration B was chosen for further experiments (refer to results section).

## MRI hard- and software

Imaging was performed on a 7 T whole-body MR system (7 T Terra, Siemens Healthineers, Erlangen, Germany). The MRI system was equipped with an 8-channel transmit and 16-channel receive (8 Tx / 16 Rx) cardiac array prototype (Rapid Biomedical, Würzburg, Germany) and tested in pTX mode with and without (identical amplitudes and phases) $B_1$-shimming. Width, length and height of the anterior and posterior housings of this array are 390mm, 350mm, and 80mm as well as 580mm, 550mm, and 50mm, respectively. Furthermore, a commercially available 1 Tx / 16 Rx coil (MRI TOOLS, Berlin, Germany) was used in sTX mode. Width, length and height of the anterior and posterior housings of this array are 346mm, 330mm, and 90mm as well as 345mm, 455mm, and 48mm, respectively. The experimental setup and coil positioning are depicted in Fig 2. For $B_1$-shimming in pTX mode, the vendor-supplied, slice-specific, automated $B_1$-shimming procedure in default mode was applied using a ROI, which included all stenosis samples. Measurements were repeated 16 times using different horizontal angles of the stenoses (15° steps, referring to the imaging plane) to assess reproducibility.

At the beginning of each experiment, a set of 2D Turbo-FLASH-localizers was used to visualize the stenosis samples in the thorax phantom. Particular effort was made to ensure that phantoms were sufficiently centered in the image to ensure the comparability of the measurements and to reduce partial volume effects. $B_0$-shimming was performed before sTX and pTX measurements, including all stenosis samples. Gradient-echo (GRE) images of the samples were acquired in coronal views of the horizontally positioned stenosis phantoms. The following acquisition parameters, taken from *in-vivo* cardiac imaging standard protocols, were used for all 2D GRE images: FOV = 256x216 mm$^2$, TR/TE = 6.2/2.37 ms, averages = 4, flip angle = 25°, acquisition time = 12 s, voxel size 0.5x0.5x4.0 mm, bandwidth 440 Hz/Px. Acquisition time was chosen short enough for scans to be comparable with *in-vivo* applications using breath-hold techniques and should be tolerated by most patients. No ECG-gating was used.

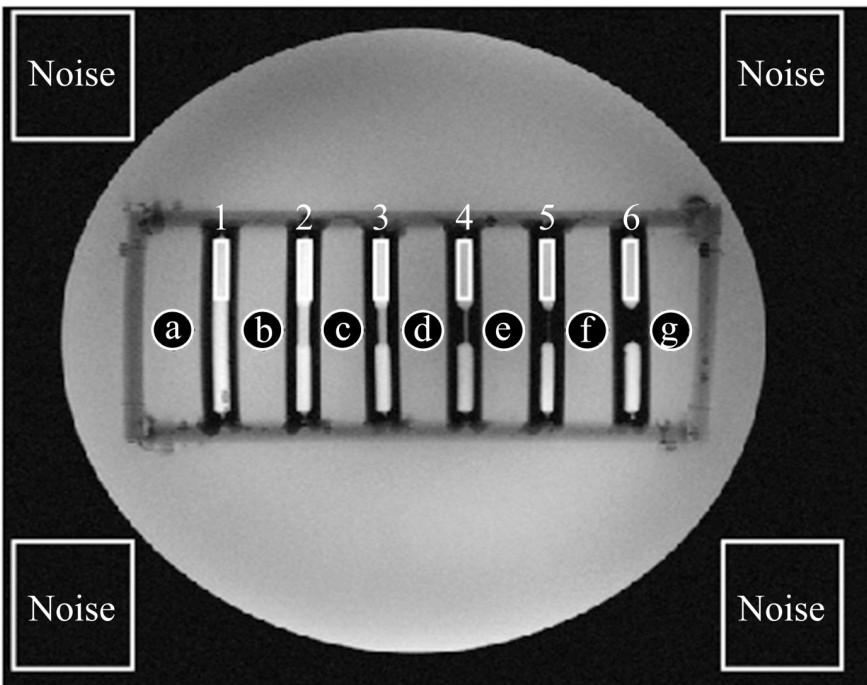

**Fig 2. Signal-to-noise ratio (SNR) evaluation.** Mean signal intensity was calculated by averaging the region of interest's (ROI) pixels' intensities (Gd-DOTA concentration 5.1 mM). Square ROIs were fitted into each of the contrast filled phantoms (1–6), circular ROIs were horizontally positioned in between the phantoms (a-g). Noise signal was drawn from the non-signal producing regions outside the phantom. $B_1$-shimming and parallel transmission yielded an 18% higher SNR inside the stenosis phantoms and 47% for outside compared to the conventional single transmit coil.

$B_1^+$-maps were measured before and after $B_1$-modification using a scanner standard turbo FLASH sequence with magnetization preparation with the following parameters: FOV = 256x216 mm$^2$, TR/TE = 12000/1.69 ms, flip angle = 10˚, voxel size 2.0x2.0x3.5 mm, bandwidth 450 Hz/Px. TR was selected so that a T1 effect was avoided.

## Image quality and stenosis evaluation

A special picture archiving and communication system (PACS) software (Merlin, Phönix-PACS, Freiburg, Germany) was used to evaluate images of this study. Two independent radiologists (A.K., reader 1 and S.H., reader 2) with seven years of experience in cardiovascular radiology evaluated this contrast agent study to assess the effects of contrast agent concentration on image quality. Readers were allowed to change window settings as needed.

All images were reviewed in a randomized and blinded order. Readers were asked to assess whether the presented image quality of each study was sufficient for diagnostic use and to rate overall image quality on a seven-point Likert scale (7 = excellent; 6 = very good; 5 = good; 4 = satisfactory; 3 = fair; 2 = poor; 1 = very poor).

Grades of stenosis were measured manually by reader 2 using standard PACS software. Grades of stenosis were determined by comparing minimal luminal diameter at the site of maximal stenosis with normal reference diameters proximal or distal, according to recommendations of the Society of Cardiovascular Computed Tomography (SCCT) [31].

## SNR evaluation

In a first step, we placed regions of interest (ROIs) in non-signal-producing areas near the corners of the image in order to evaluate the noise level. Since measurements were done with a 16-channel cardiac array, we maximized spatial distribution of these ROIs to approximate homogeneity of the noise level. As demonstrated by Constandinides et al. [32] the standard deviation of noise (σ) was approximated as the half of the root mean square value of all pixels (L) within the ROIs divided by the number of receivers used (Eq (1)). The signal was evaluated for rectangular ROIs within the phantoms and a set of seven additional circular ROIs, which were placed between the phantoms. The signal within those ROIs was calculated as the mean signal intensity ($\bar{M}$). The positioning of all ROIs is displayed in Fig 2.

SNR was calculated dividing the mean signal intensity $\bar{M}$ by the standard deviation of the noise σ (Eq (2)).

$$\sigma = \sqrt{\frac{\sum_{i=1}^{L}\left(pixel\ value\right)_i^2}{2Ln}} \tag{1}$$

$$SNR = \frac{\bar{M}}{\sigma} \tag{2}$$

Since we did not aim to characterize the hardware used, we compare and report relative values of SNR (normalized to maximum) in order to facilitate interpretation.

## Statistical analysis

Data are expressed as mean ± single standard deviation. Statistical analysis was performed with Graph Pad Prism, Version 5.0 (GraphPad Software, Inc., USA). A value of p<0.05 was considered statistically significant. Normal distribution of data was tested using the Kolmogorov-Smirnov Test. Differences in the grade of stenosis between pTX and sTX modes were tested using a Mann-Whitney *U* test for not normally distributed data. Pearson's Correlation Coefficient was used as a parametric rank statistic test to measure the strength of the association between MRI-measured and actual grade of stenosis.

# Results

## Image quality

Gd-DOTA dilution series were performed to optimize contrast settings at 7 T for stenosis visualization. Four different Gd-DOTA concentrations were filled in the stenosis phantoms. Table 1 depicts detailed results of subjective image quality ratings by two independent radiologists.

Observers found overall image quality to be very good or excellent in 100%/100% (reader 1 / reader 2) of concentration A (10.2 mM), 100%/100% of concentration B (5.1 mM), 12.5%/12.5% of concentration C (1.0 mM) and 0.0%/0.0% of concentration D (NaCl). Median Likert scores were 7.0/7.0 (reader 1 / reader 2) for concentration A, 7.0/6.0 for concentration B, 4.5/4.0 for concentration C, and 1.0/1.0 for concentration D, suggesting that concentration A and B provided equivalently sufficient image quality for stenosis evaluation. In terms of patient safety, low contrast agent concentrations are favorable. Hence, the lower concentration of the two (concentration B), was chosen for further experiments.

**Table 1. Observer ratings for Gd-DOTA dilution series.** Subjective evaluation of stenosis phantoms with different Gd-DOTA concentrations ($n$ = 16 each) using a seven-point Likert scale (7 = excellent; 6 = very good; 5 = good; 4 = satisfactory; 3 = fair; 2 = poor; 1 = very poor). Scale results are displayed as frequencies (percentages) and median values. Concentration A: 10.2 mM. Concentration B: 5.1 mM. Concentration C: 1.0 mM. Concentration D: NaCl. Reader 1 / reader 2: R1 / R2.

| Likert | Concentration A | | Concentration B | | Concentration C | | Concentration D | |
|---|---|---|---|---|---|---|---|---|
| | R1 | R2 | R1 | R2 | R1 | R2 | R1 | R2 |
| 7 | 12 (75.0) | 9 (56.3) | 10 (62.5) | 6 (37.5) | - | - | - | - |
| 6 | 4 (25.0) | 7 (43.8) | 6 (37.5) | 10 (62.5) | 2 (12.5) | 2 (12.5) | - | - |
| 5 | - | - | - | - | 6 (37.5) | 5 (31.3) | - | - |
| 4 | - | - | - | - | 8 (50.0) | 9 (56.3) | - | - |
| 3 | - | - | - | - | - | - | - | - |
| 2 | - | - | - | - | - | - | - | - |
| 1 | - | - | - | - | - | - | 16 (100.0) | 16 (100.0) |
| Median | 7.0 | 7.0 | 7.0 | 6.0 | 4.5 | 4.0 | 1.0 | 1.0 |

Subjective evaluation of stenosis phantoms with different Gd-DOTA concentrations ($n$ = 16 each) using a seven-point Likert scale (7 = excellent; 6 = very good; 5 = good; 4 = satisfactory; 3 = fair; 2 = poor; 1 = very poor). Scale results are displayed as frequencies (percentages) and median values. Concentration A: 10.2 mM. Concentration B: 5.1 mM. Concentration C: 1.0 mM. Concentration D: NaCl. Reader 1 / reader 2: R1 / R2.

## $B_1^+$-maps

$B_1^+$-maps were acquired for approximation of the field inhomogeneity. Maps with and without $B_1$-shimming are displayed in Fig 3. The map acquired with a pTX coil without $B_1$-shimming depicts distinct local $B_1^+$-inhomogeneity in the upper-right quadrant of the thorax phantom. In contrast, the $B_1^+$-map acquired with $B_1$-shimming shows a distinctly more homogenous $B_1^+$-field distribution, demonstrating the positive impact of the shimming procedure and parallel transmission.

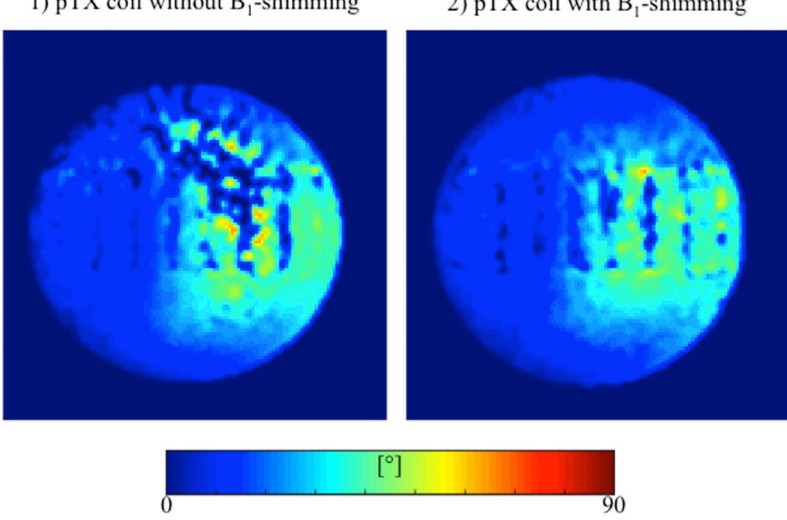

**Fig 3. Flip angle maps of stenosis phantoms with and without $B_1$-shimming.** Displayed values are limited to 90˚. Distinct flip angle inhomogeneities in parallel transmission (pTX) without $B_1$-modulation (1) are displayed as prominent dark blue shadows in the upper-right quadrant. In contrast, a much more homogenous flip angle distribution was observed using pTX with automated $B_1$-shimming (2).

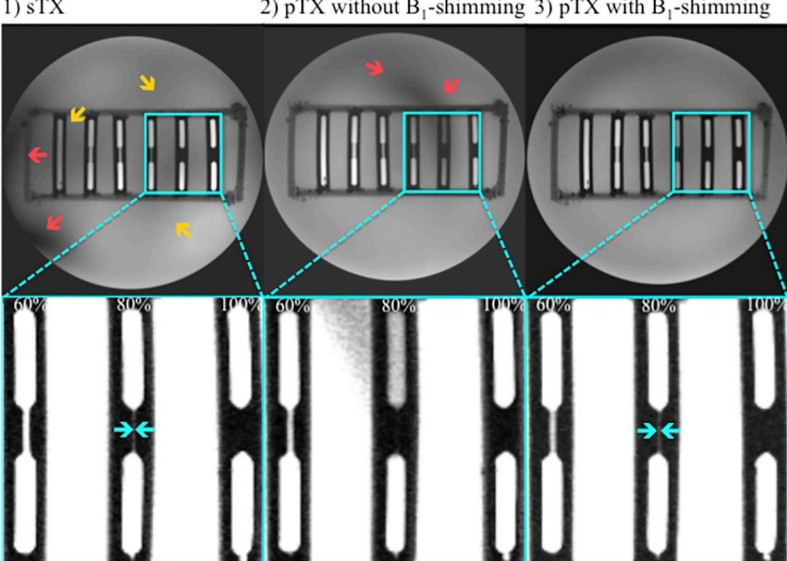

**Fig 4. Imaging of stenosis phantoms (Gd-DOTA concentration 5.1 mM).** First row: stenosis grade from left to right 0%, 20%, 40%, 60%, 80%, 100%. Second row: Magnified views of stenosis phantoms (60%, 80%, and 100%). To better visualize the stenosis area against the background, strongly windowed images were chosen for illustration. (1) Image acquired with a single transmit (sTX) coil shows strong shading artifacts only occurring at the edge of the lower-left quadrant (red arrow) and, thus, not affecting stenosis quantification. Minor field inhomogeneities in the center partly involve stenosis phantoms 0%, 60%, and 80% (yellow arrows). (2) Image from a parallel transmission (pTX) coil without $B_1$-shimming depicts strong shading artifacts in the upper-right quadrant overlaying the 80% stenosis phantom (red arrow). In this case, reasonable stenosis quantification is not possible. (3) Images acquired with a pTX coil using automated $B_1$-shimming provide superior image homogeneity and do not show relevant shading artifacts. High-grade stenoses (60% and 80%, cyan arrows) were graded significantly (p<0.01) more precise compared to the conventional coil. No significant differences in stenosis grade between pTX and sTX coils were shown for 0%, 20%, and 40%.

## MRI measurements and stenosis grading

A conventional sTX coil and a pTX RF coil with and without $B_1$-shimming were used to image stenosis phantoms (Fig 4). Results of stenosis quantification ($n$ = 15 per type of stenosis) are displayed in Figs 5 and 6. The diameter of the parent vessel of the stenosis phantoms (5 mm) was estimated significantly more precise (p<0.01) in pTX mode (mean 4.9 mm) than in sTX mode (mean 4.8 mm). Both pTX ($R^2$ = 0.99, p<0.01) and sTX ($R^2$ = 0.996, p<0.01) based stenosis evaluation showed a significantly positive correlation between actual grade of stenosis and MRI measurements. Quantification of stenoses using the sTX coil slightly overestimated high-grade stenoses by up to 6.1±1.8% (Table 2).

Visualization of stenoses using the pTX coil with $B_1$-shimming was more precise than using the sTX coil. Furthermore, the deviation from the true stenosis diameter was significantly lower for moderate to high-grade stenoses (Fig 6, 60%: p<0.01; 80% p<0.01). In images acquired with the pTX coil without $B_1$-shimming, stenosis grading, especially of high-grade stenoses, was not reasonably possible due to strong shading artifacts caused by severe interfering $B_1^+$-inhomogeneities (Fig 4). In the setting of this study, this especially affected high-grade stenoses in the upper-left quadrant.

## SNR evaluation

Mean noise levels of the four ROIs in the image corners deviated less than 1% from the mean value of all ROIs. Across the thorax phantom, pTX and $B_1$-shimming yielded an increase of

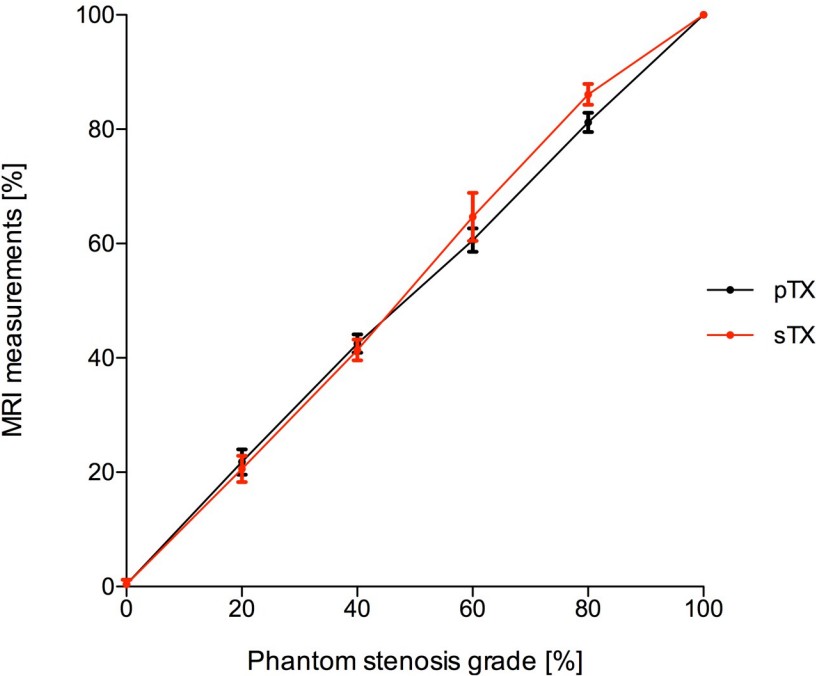

**Fig 5. Quantification of stenosis grades in 3D-printed phantom models using a conventional single transmit coil (sTX, red dots) and a parallel transmission (pTX) coil with $B_1$-shimming (black dots).** Stenosis grades were determined by comparing the minimal luminal diameter at the site of maximal stenosis with normal reference diameters proximal or distal. Error bars denote single standard deviation. In images acquired with the conventional coil, higher grade stenoses were slightly overestimated, whereas pTX with $B_1$-shimming enabled significantly (p < 0.01) more precise stenosis quantification. Quantification of stenoses in images of pTX coil without $B_1$-shimming was not reasonably achievable in high-grade stenoses due to severe $B_1^+$-inhomogeneities.

14% in SNR for the blood pool (inside the stenosis phantoms) and 32% for the surrounding tissue (outside the phantoms), with respect to non-manipulated RF fields. Results of the SNR evaluation (relative SNR and standard deviation) are summarized in Table 3 (inside the phantoms) and Table 4 (outside the phantoms).

## Discussion

This study indicates that $B_1$-shimming and pTX can distinctly improve image homogeneity and, thus, be beneficial for evaluating coronary artery stenosis phantoms. In gradient-echo sequences, $B_1$-shimming markedly reduced shading artifacts caused by local inhomogeneities of the $B_1^+$-field enabling a more balanced and precise stenosis evaluation. Imaging with a conventional sTX coil and a pTX coil without $B_1$-shimming showed more shading artifacts. In severe cases, when local signal drops coincided with the location of the stenoses, quantification was not reasonably possible. In images acquired with a pTX coil using $B_1$-shimming higher grade stenoses were rated significantly more accurately than when using a commercial sTX coil. In mild to moderate grade stenoses, both sTX and pTX approaches did not show significant differences.

Efficient visualization and quantification of vascular stenoses require a combination of high SNR and $B_1^+$-field homogeneity, which is associated with a homogenous flip angle distribution. Low flip angles may lead to a local signal reduction, failure of magnetization preparation pulses, and eventually to artifacts and biased quantitative measures [25]. $B_1^+$-field

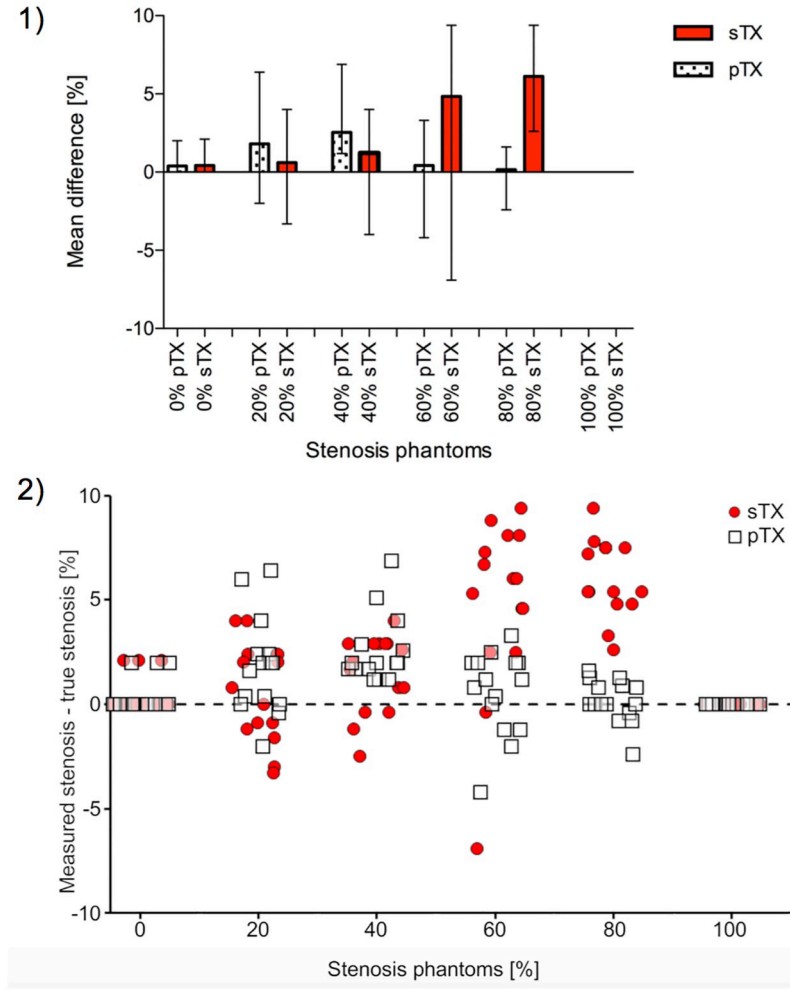

**Fig 6. Measurement accuracy in stenosis quantification using a conventional single transmit coil (sTX) and a parallel transmission (pTX) coil with B₁-shimming.** (1) For higher-grade stenoses, the mean differences obtained by subtracting real measures of the 3D-printed phantoms from MRI measurements were significantly lower (60% and 80%: $p<0.01$) using pTX with $B_1$-shimming. Error bars indicate the range of the differences across the phantoms. (2) Deviation from actual stenosis determined with pTX (white squares) and sTX (red circles). To visualize the individual data points, the values on the x-axis (true degree of stenosis) are not fixed to a single value (e.g. 20%), but have a range of +/-5% (e.g. 15% to 25%). The y-axis indicates the difference between the stenosis measured with the respective technique and the true degree of stenosis.

**Table 2. Results of stenosis diameter analysis [mm] using single transmission (sTX) and parallel transmission (pTX) imaging (*n* = 15 per type of stenosis).** Data are mean ± standard deviation.

| stenosis (%) | 0 | 20 | 40 | 60 | 80 | 100 |
|---|---|---|---|---|---|---|
| pTX coil (with $B_1$-shimming, %) | 0.4 ± 0.8 | 21.8 ± 2.2 | 42.5 ± 1.6 | 60.6 ± 2.1 | 81.2 ± 1.7 | 100 ± 0 |
| conventional coil (sTX without $B_1$-shimming, %) | 0.4 ± 0.8 | 20.6 ± 2.3 | 41.4 ± 1.8 | 64.7 ± 4.2 | 86.1 ± 1.8 | 100 ± 0 |

Results of stenosis diameter analysis [mm] using single transmission (sTX) and parallel transmission (pTX) imaging (*n* = 15 per type of stenosis). Data are mean ± standard deviation.

**Table 3. Signal-to-noise ratio (SNR) measurements in the stenosis phantoms (blood pool).** Data are mean ± standard deviation.

|      | ROI 1 | ROI 2 | ROI 3 | ROI 4 | ROI 5 | ROI 6 | Median |
|------|-------|-------|-------|-------|-------|-------|--------|
| pTX  | 0.89±0.12 | 1.00±0.12 | 0.87±0.14 | 0.67±0.11 | 0.71±0.13 | 0.14 | 0.81±0.13 |
| sTX  | 0.40±0.12 | 0.59±0.12 | 0.71±0.14 | 0.68±0.11 | 0.69±0.13 | 0.84±0.14 | 0.67±0.13 |

Data are mean ± standard deviation.

homogeneity is dependent on different prerequisites of the applied coil and the imaged object. The visibility of stenoses depends on the signal of the stenosis vs. the surrounding variation of signal intensities on a pixel-by-pixel basis. Thus, the reason for the superior stenosis evaluation with pTX and $B_1^+$-shimming possibly lies in reduced $B_1^+$-inhomogeneity because local variations of $B_1^+$-induced signal variations may diminish the visibility of the stenosis. However, this finding may also be caused by overall higher SNR values compared to the conventional coil. Other reasons might be image errors, e.g. based on slice selection, in areas of inhomogeneous $B_1^+$ or a combination of these mechanisms. In the context of this study, the underlying causes could not be fully identified and require further in-depth analysis.

In this study, two observers rated images of stenosis phantoms acquired with different Gd-DOTA concentrations typical for human studies. Both full and half concentration of a conventional Gd-DOTA dose for humans were equally suitable for stenosis evaluation. These results suggest that lower gadolinium concentrations than customary at 1.5 and 3 T may be sufficient for stenosis grading at UHF-CMR. In terms of patient safety, the application of reduced contrast agent concentrations without loss of image information is beneficial [33–35]. These findings are in concordance with previously published brain UHF-MRI studies [6, 36].

The results from this study also indicate that local $B_1^+$-inhomogeneities may severely impair stenosis evaluation under certain conditions. In a phantom study, local signal drops may be corrected manually by repositioning of the imaged objects, adjusting the load of the coil, or changing imaging parameters. Under *in-vivo* conditions, however, this approach is hampered since it is hardly possible to detect slight shading artifacts overlaying small coronary vessels in the complexity of the human heart. Furthermore, the position of signal drops cannot be predicted reliably. Automated $B_1$-shimming and pTX might overcome these constraints to some degree and improve imaging of small vessels such as coronary arteries. Future studies should further address challenges with $B_0$- and $B_1^+$-inhomogeneity at 7 T to take advantage of increased SNR at ultra-high fields. This approach may help to enhance the diagnostic value of CMR and its current limitations at 1.5 and 3 T, since higher SNR may be used for improved spatial resolution or shorter scan times and thus reduce influence of motion.

The imaging parameters and phantom dimensions in this study were selected to be as close as possible to a human application. To enhance transferability, acquisition time was chosen to be similar to *in-vivo* measurements using breath-hold techniques. However, *in-vivo* conditions differ substantially from our thorax phantom due to overall SNR levels as well as susceptibility changes between different tissues, and motion artifacts. Hence, for a transfer to a human

**Table 4. Mean signal-to-noise ratio (SNR) values and standard deviation in the thorax phantom of the surrounding tissue (outside the stenosis phantoms).**

|      | ROI a | ROI b | ROI c | ROI d | ROI e | ROI f | ROI g | Median |
|------|-------|-------|-------|-------|-------|-------|-------|--------|
| pTX  | 0.88±0.04 | 1.00±0.03 | 0.90±0.05 | 0.75±0.04 | 0.76±0.03 | 0.89±0.04 | 0.84±0.05 | 0.88±0.04 |
| sTX  | 0.39±0.06 | 0.45±0.04 | 0.56±0.04 | 0.57±0.05 | 0.50±0.05 | 0.58±0.03 | 0.40±0.04 | 0.56±0.05 |

The ROI selection is displayed in Fig 2. Overall, SNR was distinctly higher with $B_1$-shimming and pTX than with the conventional sTX coil.

system several things such as varying breath hold capability, $B_0$-shimming at 7 T or quality of the gating signal ECG, would have to be considered and the sequence would have to be adapted and reevaluated.

From a clinical perspective, it is not only the diameter of a stenosis that plays a key role in the assessment of coronary artery disease. Other essential factors are length, inner structure, the opening angle of a stenosis, and the constitution of the stenosis, which forms swirls and turbulences that hamper blood flow. Although statistical significance was reached when comparing moderate and severe stenosis grades with sTX and pTX coils, this is unlikely to be clinically relevant. The decision on further management would not be based on a few percentage points. A lesion would still be classified as 'significant' and would therefore be treated appropriately. However, it is potentially clinically relevant, if high-grade stenoses are misinterpreted as low grade stenoses.

As a limitation of this study, a performance evaluation of the pTX coil with and without $B_1$-shimming was not reasonably achievable in high-grade stenoses due to severe interfering $B_1^+$-inhomogeneities observed when imaging without $B_1$-modification. Moreover, image noise analysis based on ROIs placed near the corners of the image may be biased in multi-channel receive coils [37]. In this study, we approximated SNR using the standard deviation of noise $\sigma$ and the measured mean signal intensity $\bar{M}$. SNR values in this study, where $\bar{M}/\sigma > 10$ is true in all cases, are therefore subject to errors $\leq 5\%$, as demonstrated by Constantinides et al. [32]. Differences in the noise level were negligible comparing non-signal producing ROIs in the four corners of our images. The approach for SNR estimation by Constantinides et al. assumes the absence of noise correlation. This is unlikely to be the case for 16-channel receive coils. We therefore used relative SNR in this study, which additionally eases direct comparison. Bias in the SNR and noise most likely did not influence the results and the validity of the conclusions. A detailed comparison of the two different RF coil types (conventional sTX and prototype pTX coil) was not part of this study. Instead, our proof of concept study focused on the quantifiable effects of $B_1$-shimming on image homogeneity. The partial volume effect can introduce significant errors in quantitative measurements, which plays an important role when imaging delicate structures such as high-grade stenoses in small vessels. The relatively large diameter of the reference vessel (5mm) limits applicability to coronary artery lesions that involve smaller major epicardials. Differences in phantom angulation might have hampered stenosis evaluation. However, much care was taken to ensure comparable imaging conditions between the different imaging modes, such as precise angulations of the phantoms. In stenosis phantoms, the distribution of the contrast agent was uniform, and slices were aligned to the centerline of the stenosis phantoms. These assumptions do not hold for imaging of the human vasculature. For example, atherosclerotic lesions in patients are not homogeneous and are frequently eccentric, which may impact diagnostic accuracy. The effects of flow and tissue inhomogeneity in human systems were not part of this study.

The phantom setup used in this study is relatively simple. Hence, direct translation of observed $B_1$-inhomogeneity to the human body is limited. However, at 7T the $B_1$-profiles in the thorax vary significantly with the subject weight, thorax shape, body-mass-index and fat-muscle distribution and no single phantom however complex could accurately represent this variety. We believe that the rather simple setup can already provide $B_1$-conditions relevant for imaging of vascular stenosis, in particular considering the aim of this study: to assess capability and limitation of the vendor-provided shimming platform using a pTX-array. In addition to cardiac MRI the coils used in this study may be also applicable to imaging other locations e.g the vasculature of the lower extremities. Thus, results of this study may be applicable to such imaging applications as well, indicating that the vendor-integrated shimming process may

already improve imaging conditions with respect to $B_1$-inhomogeneity and therefore, also improve stenosis visualization in non-cardiac vessels.

Although RF shimming is a promising technique, it also has its limitations [38]. In particular, it requires a robust measurement of $B_1$-field maps of individual transmit channels. Currently, the vendor-integrated $B_1$-calibration procedure takes about 40 sec in non-triggered mode, but elongates to 2 minutes by activation of cardiac gating, which would be a requirement for accurate $B_1$-field maps. This is not compatible with breath-hold measurements and research developments of calibration methods is required [39, 40]. Moreover, currently, there is no vendor-provided subject-specific SAR safety concept available for the application of pTX-based $B_1$-shimming (both static and dynamic) for thorax pTX-arrays in human subjects. Commercially available pTX arrays running in pTX mode are thus currently using preset phase and amplitude values, which have previously been validated with respect to SAR safety. This means that $B_1$-shimming settings are computed for generic electromagnetic body models („Duke", „Ella"[SH1] etc) and then fixed for a specific coil. Operating with $B_1^+$-shimming in 7 T thorax MRI, in general, requires the availability of highly experienced personnel and infrastructure for the electromagnetic simulations ensuring the SAR safety. To date, $B_1$-shimming is a non-standard technique, where ongoing research may significantly improve image quality in future UHF cardiac MRI.

## Conclusion

$B_1$-shimming and pTX can improve overall image homogeneity at 7 T, which is especially beneficial when evaluating small vessel structures such as coronary arteries. Therefore, $B_1$-shimming might play an essential role in the future of UHF-CMR.

## Supporting information

**S1 Data. Original data.**
(XLSX)

## Acknowledgments

This publication was supported by the Open Access Publication Fund of the University of Wuerzburg.

## Author Contributions

**Conceptualization:** Stefan Herz, Maria R. Stefanescu, Maxim Terekhov, Andreas M. Weng.

**Data curation:** Stefan Herz, Maria R. Stefanescu, David Lohr, Aleksander Kosmala, Jan-Peter Grunz.

**Formal analysis:** Stefan Herz, Maria R. Stefanescu, David Lohr, Aleksander Kosmala, Maxim Terekhov.

**Funding acquisition:** Laura M. Schreiber.

**Investigation:** Stefan Herz, Maria R. Stefanescu, David Lohr, Aleksander Kosmala, Jan-Peter Grunz.

**Methodology:** David Lohr, Patrick Vogel, Maxim Terekhov.

**Project administration:** Stefan Herz, Maria R. Stefanescu, Thorsten A. Bley, Laura M. Schreiber.

**Resources:** Patrick Vogel, Thorsten A. Bley, Laura M. Schreiber.

**Supervision:** Maxim Terekhov, Thorsten A. Bley, Laura M. Schreiber.

**Validation:** Stefan Herz, David Lohr, Patrick Vogel, Maxim Terekhov, Andreas M. Weng, Thorsten A. Bley, Laura M. Schreiber.

**Visualization:** Stefan Herz, Maria R. Stefanescu, David Lohr.

**Writing – original draft:** Stefan Herz.

**Writing – review & editing:** Stefan Herz, Maria R. Stefanescu, David Lohr, Patrick Vogel, Aleksander Kosmala, Maxim Terekhov, Andreas M. Weng, Jan-Peter Grunz, Thorsten A. Bley, Laura M. Schreiber.

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
