## [Decision Letter · Decision Letter 0]

22 Dec 2021

PONE-D-21-01916Effects of Image Homogeneity on Stenosis Visualization at 7 T in a Coronary Artery Phantom Study: With and without

B1-Shimming and Parallel TransmissionPLOS ONE

Dear Dr. Herz,

Thank you for submitting your manuscript to PLOS ONE. After careful consideration, we feel that it has merit but does not fully meet PLOS ONE’s publication criteria as it currently stands. Therefore, we invite you to submit a revised version of the manuscript that addresses the points raised during the review process.

The manuscript has been evaluated by two reviewers, and their comments are available below.

The reviewers have raised a number of concerns that need attention. They request additional information on methodological aspects of the study. In particular some of these points relate to omissions in the manuscript including the rationale for methodological choices, supporting literature/points which may enable results to support conclusions, and more descriptive materials/methods which detail equipment and protocols employed. Please note that whilst there are comments in the reviews regarding what new information is added by the manuscript, PLOS ONE does not judge manuscripts on perceived novelty, but rather on methodological rigor. It is highly recommended therefore to address the methodological concerns raised by the reviewers. In addition whilst we do not judge manuscripts for perceived novelty, it is recommended to address the reviewer's concerns regarding the rationale for experimental choices and the context for this work.

Could you please revise the manuscript to carefully address the concerns raised?

We look forward to receiving your revised manuscript.

Kind regards,

Sebastian Shepherd

Associate Editor

PLOS ONE

Journal Requirements:

Reviewers' comments:

Reviewer's Responses to Questions

**Comments to the Author**

1. Is the manuscript technically sound, and do the data support the conclusions?

Reviewer #1: Yes

Reviewer #2: No

2. Has the statistical analysis been performed appropriately and rigorously? 

Reviewer #1: Yes

Reviewer #2: I Don't Know

3. Have the authors made all data underlying the findings in their manuscript fully available?

Reviewer #1: No

Reviewer #2: Yes

4. Is the manuscript presented in an intelligible fashion and written in standard English?

Reviewer #1: Yes

Reviewer #2: Yes

5. Review Comments to the Author

Reviewer #1: In this article, the authors investigated the effects of B1-shimming pTX on image homogeneity compared with pTX without B1-shimming on ultra-high field cardiac magnetic resonance imaging, they compared the measurements of stenosis using phantoms filled with gadolinium-based contrast agent (for Gd-DOTA) performed on images acquired with B1-shimming pTX with those obtained from images acquired with commercial sTX, they rated qualitatively the subjective evaluation of the stenosis for different Gd-DOTA dilution series, and they compared SNR measurements on images acquired with B1-shimming pTX compared with sTX. I believe this study is interesting to the field and has been performed elegantly and properly. The research objectives are well-defined, and the study design is appropriate. The figures and tables summarizing the main results are good. The article is well-written and its structure and organization are also good. The discussion section is informative and are contextualized with the results from peer-reviewed articles relevant to the topic. The references are in general appropriate, and include both newer and older studies. The conclusions drawn by the authors are in general supported by the data they present.

I have only a few minor points:

1. The authors should clarify how many measurements per type of stenosis have been performed by reader 2.

2. Similarly, in figure 6, it will probably be more informative to show all the data points in addition to the mean and standard deviations of the bar plots.

3. “Although RF shimming is a promising technique, it also has its limitations [36].” It would be interesting to list a few of them.

Reviewer #2: General comment:

In this study, the authors investigate the impact of B1-shimming on image homogeneity and stenosis evaluation at 7T in a phantom filled with with gadolinium based contrast agent. They also compare these results with those obtained with a single channel transmit coil.

While in principle the long term target is relevant in terms of public health (coronary stenosis evaluation with MRI at 7T), this study unfortunately does not bring new knowledge, and suffers from serious methodological shortcomings preventing the work from having meaningful relevance to real in-vivo situation. Here are a few examples of such deficiencies:

- the authors chose a static phantom, whereas blood flow is a fundamental and critical characteristic of MR angiography.

- multiple studies have carefully described ways to address the very challenging B0 and B1 inhomogeneities occurring in the torso at 7T. Here, the chosen phantom is not representative of real torso anatomy, whereas it has been described elsewhere, and in details, how B1 shimming and B0 shimming solutions need to be specifically optimized to their target in cardiovascular imaging at 7 Tesla.

- the "non-B1 shimmed" setting taken as a comparison basis here seems fairly meaningless: it is very well known that at 7T B1 shimming are needed to address B1 inhomogeneities and several approaches have been proposed for these challenges. Here, the authors simply rely on the vendor-based B1 mapping and shimming technique without reporting any details on parametric quantities. The "non B1 shimmed" setting seems to be defined arbitrarily at best. RF efficiency and related SAR impact are not discussed. All these considerations can be critical for a technique meant to be ultimately applied in humans.

- this study does not include electromagnetic simulations that have become almost unavoidable when addressing transmit B1 issues with multiple channels at 7T.

- there is no description of the single transmit RF coil (not even the size is reported)

Even before considering the aforementioned problems, the purpose of the study seems quite elusive from the beginning. Indeed, in experiments like the ones reported by the authors, it is perfectly known that poor B1 profiles will occur (non shimmed multi channel transmit coils at 7T) and that the latter will translate in very poor and heterogeneous Signal to Noise Ratio (SNR) (multiple studies have reported this in details). Observing poorer precision in diameter or stenosis evaluation when SNR is low and heterogeneous is a straightforward and fully expected consequence. That transmit B1 contributes to this deleterious effect is as much predictable. Likewise, the comparison between the impact of single or multi channel transmit coil does not tell more than what is already known about transmit B1 characteristics.

As a result, this reviewer could not identity relevant conclusions coming from the study.

Specific comments:

1. Page 4, line 7. MRI is not a radiation-free imaging technique as claimed by the authors. It is free of ionizing radiation, but electromagnetic waves are radiations, even at radio frequencies. This language needs to be modified.

6. PLOS authors have the option to publish the peer review history of their article (what does this mean?). If published, this will include your full peer review and any attached files.

Reviewer #1: **Yes: **Giuseppe Barisano

Reviewer #2: No

---

## [Author Response · Author response to Decision Letter 0]

19 Feb 2022

Please find a point-by-point response to the reviewers comments enclosed.

---

## [Decision Letter · Decision Letter 1]

30 Mar 2022

PONE-D-21-01916R1

Effects of Image Homogeneity on Stenosis Visualization at 7 T in a Coronary Artery Phantom Study: With and without

B1-Shimming and Parallel Transmission

PLOS ONE

Dear Dr. Herz,

Thank you for submitting your manuscript to PLOS ONE. After careful consideration, we feel that it has merit but does not fully meet PLOS ONE’s publication criteria as it currently stands. Therefore, we invite you to submit a revised version of the manuscript that addresses the points raised during the review process.

I acknowledge that one of the previous reviews was my own.

The manuscript has now been re-evaluated by myself and the other former reviewer, whose comments are available below.

While the reviewer notes that the relevance of the work is still questionable, the reviewer agrees with the content of the additions provided by the authors in this review and gives credit to the authors to have further emphasized the challenges and limitations of 7T cardiac imaging as well as the expected differences between their phantom model and in vivo acquisitions.

PLOS ONE does not judge manuscripts on perceived novelty and impact, but rather on methodological rigor. The rationale for experimental choices and the context for this work have been satisfactorily provided by the authors.

Please note the minor typo found by the reviewer, that needs to be corrected in your revised submission.

I appreciate the authors adding the graphs with all data points in figure 6, which add transparency to the actual analysis performed. I believe these additional graphs are important and should be included in figure 6. However, I have several concerns related to these graphs and in general to figure 6:

Please make sure that all data points are visible in the additional graphs on panel b and c: there might be some overlapping points which prevent them to be visible (see for example for stenosis type 0%, only 1 point is visible, both in panel b and in panel c, instead of the stated 15 points).The y-axis name “mean absolute difference” in panels b and c is possibly incorrect, since negative values are also present.The legend for panel a currently reads “Error bars indicate the range of the absolute differences across the phantoms”, which seems incorrect since, also in this case, the error bars include negative values.The error bars in panel a are confusing because they look to be the opposite of the actual data shown in panels b and c: if I understand this correctly, the top of the error bar represents more negative differences, whereas the bottom of the error bar represents more positive differences. My recommendation is to report the mean difference in panel a, rather than the mean absolute difference. This would also facilitate the interpretation, showing that there is overestimation of the MRI measurement compared with real measurement rather than underestimation (which might not be obvious to all the readers). If the authors want to follow this recommendation, please edit the y-axis and the text, and in the figure legend please specify whether the difference is “MRI – real” or “real – MRI”.If the authors want to keep the mean absolute difference, please specify in the figure legend the sign of the error bars.In the text, the authors wrote: “Quantification of stenoses using the sTX coil slightly overestimated high-grade stenoses by up to 6.1±1.8%”. Therefore, I suppose that the differences (not absolute) in panels b and c are obtained by subtracting the real measure from the MRI measurement. Please clarify this point in the legend of figure 6 panels b and c. Could you please revise the manuscript to carefully address these points?

We look forward to receiving your revised manuscript.

Kind regards,

Giuseppe Barisano, M.D.

Academic Editor

PLOS ONE

Journal Requirements:

Reviewers' comments:

Reviewer's Responses to Questions

**Comments to the Author**

1. If the authors have adequately addressed your comments raised in a previous round of review and you feel that this manuscript is now acceptable for publication, you may indicate that here to bypass the “Comments to the Author” section, enter your conflict of interest statement in the “Confidential to Editor” section, and submit your "Accept" recommendation.

Reviewer #2: (No Response)

2. Is the manuscript technically sound, and do the data support the conclusions?

Reviewer #2: Partly

3. Has the statistical analysis been performed appropriately and rigorously? 

Reviewer #2: I Don't Know

4. Have the authors made all data underlying the findings in their manuscript fully available?

Reviewer #2: Yes

5. Is the manuscript presented in an intelligible fashion and written in standard English?

Reviewer #2: Yes

6. Review Comments to the Author

Reviewer #2: General comment.

In this revision the authors included a number of comments and clarifications aiming at addressing this reviewer's comments. While agreeing with the content of these additions, this reviewer is still confronted to a dilemma.

On the one hand, while the results obtained by the authors do not seem to be technically incorrect, what really is at stake is the relevance of the work. The authors refer to a 2021 study published in PlosOne on cardiac MRI at 7T in a cohort of 84 patients, but in this study the focus is not on MR angiography and the assignement of myocard segments to coronary artery branches is solely based on morphometry. The same paper illustrates in one figure the visibility of the proximal right coronary and of the proximal left coronary, but this is reported in a rather anecdotical fashion. To this day, the very small number of publications (not even an handful) that have been dedicated to visualizing coronary arteries with MRA at 7T are dated back to 2014 or before, which reflects upon the technical challenges still faced for cardiac imaging at 7T.

In their introduction, the authors insist on the expected advantages of 7T (higher SNR and parallel imaging performances, potentially better contrast) and they emphasize the superiority of 7T over 1.5T and 3T in this regard, but it cannot be claimed that the difficulties faced to addressed B0 , B1 and SAR related issues at 7T in the heart have been remedied in a suitable manner to consistently provide satisfactory quality.

On the other hand, the reviewer gives credit to the authors to have, in this revision, further emphasized the challenges and limitations of 7T cardiac imaging as well as the expected differences between their phantom model and in vivo acquisitions.

Specific point.

PDF page 4, before-last line, typo. The last sentence on this page starts with "1However..." instead of "However..."

7. PLOS authors have the option to publish the peer review history of their article (what does this mean?). If published, this will include your full peer review and any attached files.

Reviewer #2: No

---

## [Author Response · Author response to Decision Letter 1]

8 Jun 2022

Please refer to the rebuttal letter.

---

## [Editor Report · Decision Letter 2]

16 Jun 2022

Effects of Image Homogeneity on Stenosis Visualization at 7 T in a Coronary Artery Phantom Study: With and without B1-Shimming and Parallel Transmission

PONE-D-21-01916R2

Dear Dr. Herz,

We’re pleased to inform you that your manuscript has been judged scientifically suitable for publication and will be formally accepted for publication once it meets all outstanding technical requirements.

Kind regards,

Giuseppe Barisano, M.D.

Guest Editor

PLOS ONE

---

## [Editor Report · Acceptance letter]

20 Jun 2022

PONE-D-21-01916R2 

Effects of Image Homogeneity on Stenosis Visualization at 7 T in a Coronary Artery Phantom Study: With and without B1-Shimming and Parallel Transmission 

Dear Dr. Herz:

I'm pleased to inform you that your manuscript has been deemed suitable for publication in PLOS ONE. Congratulations! Your manuscript is now with our production department. 

Kind regards, 

on behalf of

Dr. Giuseppe Barisano 

Guest Editor

PLOS ONE